# The Characteristics of the Reduction of Interference Effect during Dual-Task Cognitive-Motor Training Compared to a Single Task Cognitive and Motor Training in Elderly: A Randomized Controlled Trial

**DOI:** 10.3390/ijerph20021477

**Published:** 2023-01-13

**Authors:** Justyna Wiśniowska, Emilia Łojek, Agnieszka Olejnik, Anna Chabuda

**Affiliations:** 1Faculty of Psychology, University of Warsaw, 00-183 Warsaw, Poland; 2Faculty of Physics, University of Warsaw, 02-093 Warsaw, Poland

**Keywords:** ageing, dual-task training, interference effect

## Abstract

Many studies have indicated a weakening in several areas of cognitive functioning associated with the normal ageing process. One of the methods supporting cognitive functions in older adults is dual-task training which is based on performing cognitive and motor exercises at the same time. The study aimed at examining the characteristics of dual-task training compared to single-task training in participants over 65 years of age. Sixty-five subjects took part in the study. They were randomly assigned to three groups: dual-task cognitive-motor training (CM), single-task cognitive training (CT), and single-task motor training (MT). The training program in all groups encompassed 4 weeks and consisted of three, 30-min meetings a week. Specialized software was designed for the study. The main indicators, such as orientation and planning time and the number of errors, were monitored during the whole training in all groups. The obtained results have shown that the dual-task training was associated with a significantly greater number of movement errors, but not with a longer task planning time compared to the single-task condition training. There was a decrease in the time needed to plan a path in the mazes by subjects training in the CM, CT, and MT groups. The results indicate that after each type of training, the number of errors and the time needed to plan the path decrease, despite the increasing difficulty of the tasks. The length of planning time was strongly correlated with the number of errors made by individuals in the CM group (r = 0.74, *p* = 0.04), compared to the ST group—for which the said correlation was not significant (r = 0.7, *p* = 0.06). The dual-task cognitive-motor training is more cognitively demanding compared to the single-task cognitive and motor training. It manifests in a greater number of errors, but it does not extend the orientation and planning time.

## 1. Introduction

Cognitive, especially executive, and motor functions, which tend to decline with age, are important in everyday life as the performance of some of the most basic activities requires the simultaneous involvement of both types of these functions [1,2,3]. Performing two activities at the same time (dual-task—DT) can pose a significant burden for executive functions (EF), especially in the elderly. One of the most important aspects of this mechanism is the interference effect (IE).

The interference effect is defined as overlapping of similar pieces of information, which may inhibit or disrupt one another, thus making it easier to forget one or both [4]. Most often, the IE is explained on the basis of the limited resources [5,6] or bottleneck theories [7,8]. According to the first one, several tasks executed at the same time can exceed the capacity of cognitive resources [5,6]. These tasks can interfere with each other, causing the inhibition of highly automated operations over the less intuitive ones. The above phenomenon has long been known from research on the Stroop effect [9]. The completion time of dual-task, in comparison to the single-task, increases and more errors may appear; moreover, considerable attention resources and memory, thinking, planning, which are components of EF, are also used [5,6,9]. In turn, the bottleneck theory explains that attention is not an unlimited resource. The stimuli in the brain are filtered in such a way that only the most important ones reach it. Consequently, during DT, the priority will be set to one of the tasks. The interference is related to the effect of the time needed to decide which task should be performed first [7,8].

Research conducted on the basis of both theories shows that the IE may change and increase if the task is difficult, or it may be reduced in the course of the training [9,10,11,12]. Davidson, Zacks, and Williams [13] have shown that the reduction of the IE also occurs in the elderly compared to younger participants in the Stroop tasks. The elderly exhibited a greater interference effect than the younger subjects throughout the training. Older subjects made more mistakes, and it took them longer to complete the tasks. Importantly, however, in both groups, the effect of reduced interference during practice was observed, and the older adults showed the same trend of improvement as the younger participants. The authors concluded that the reduction of the interference effect during practice is present regardless of age. Similar results were also found in later studies [13,14,15,16,17]. In the Burger et al. [14] study, the elderly and younger subjects participated in 5-day training based on the Stroop tasks. The results revealed the significance of differences in the learning process in younger and elderly participants. The younger subjects generally showed a reduction of the interference effect at the final stage of training, whereas in the elderly, the reduction of the interference effect occurred regularly throughout the training. The number of years of formal education of the subjects played the most important role in predicting the benefits achieved in training. The greatest effect of reducing interference was observed in older participants with higher education.

The IE reduction was also investigated in studies applying neurophysiological tests, using functional magnetic resonance (fMRI), electroencephalography (EEG), and functional near-infrared spectroscopy (fNIRS) in participants between 20 and 30 years old [18]. In the fMRI study, four different patterns of brain activity were distinguished during execution of cognitive-motor dual-tasks [19]. In the EEG study, the reduction of the P300 amplitude during DT execution compared to a single task was obtained [20,21]. The fNIRS study showed that the level of oxygenation in the prefrontal cortex increases during DT compared to a single task [22,23].

In summary, performing two activities simultaneously can overload a person’s cognitive resources, causing an interference effect. However, the IE may be reduced by multitasking training. As shown by contemporary works [14], the reduction of interference effect turns out to be more significant in seniors compared to young individuals. The reduction of the IE is also evident in neurophysiological tests, in studies with fMRI, EEG, and fNIRS.

The cognitive mechanisms underlying the performance of two activities at the same time may explain supporting the elderly through the dual-task training. Unfortunately, the mechanism of reducing the interference effect has not been fully understood yet. Referring to the above-mentioned literature on the subject, the assumption is that the dual-task training is more cognitively demanding compared to the single-task training, consequently:

The planning time will be longer, and the number of errors will be larger during the dual-task training, compared to the single-task training.

The planning time will be shorter, and the number of errors will be reduced after all types of training.

The main goal of this study is to answer the question of how the characteristics of dual-task training may be compared to the single-task training in the elderly.

## 2. Materials and Methods

### 2.1. Participants

Sixty-five participants (16 men and 48 women), over 65 years of age, without neurological, psychiatric, cardiological, or orthopedic diseases/disorders, moving independently, were recruited from senior community centers in Warsaw. Exclusion criteria were a score below 24 points in the Polish version of the Mini-Mental State Examination test (the Polish adaptation and standardization of MMSE [23]), no vision correction and assisted walking.

The total number of screened participants comprised 84 subjects. The participants of the study were randomly assigned to one of the three experimental groups and one control group (4 men, 16 women). Further, 58 of them completed the training (14 men and 44 women): 19 participants (5 men, 14 women) in the cognitive-motor group (4 individuals resigned), 20 (3 men, 17 women) in the cognitive group, 19 (6 men, 13 women) in the motor group (2 individuals dropped out). Six individuals dropped-out of the experimental groups due to the length of training. Only data obtained from the participants who completed the study were included in the statistical analysis. There were no training data from the control group.

The mean age of the study participants was 71.2 (SD = 5.2), the number of years of education—15.3 (SD = 2.7), and the mean MMSE score was 29.1 (SD = 0.9). There were no demographic and clinical (MMSE score) differences between experimental and control groups. Detailed data are presented in Table 1. All participants agreed not to engage in any form of rehabilitation during the study period.

All patients provided informed consent prior to inclusion. The research was completed in accordance with Helsinki Declaration.

#### Randomization

Each examined person, after receiving a telephone notification, was randomly assigned by the project manager to one of four groups: Cognitive Motor Dual-Task Group (CM), Cognitive Task Group (CT), Motor Task Group (MT), or Control Group (C), and obtained a consecutive number from 1 to 20, in accordance with the simple randomization methodology [19]. Random assignment to groups was carried out using Microsoft Excel 2017 (365 Personal version).

### 2.2. Procedure

The participants were randomly assigned to one of three groups. The training in the CM group consisted in performing cognitive-motor tasks, the training in the CT group consisted in performing cognitive tasks, and the training in the MT group consisted in performing motor tasks. In the control group, the participants underwent two neuropsychological assessments (especially focusing on executive functions and attention), identical to the pre-test and post-test for the experimental groups, as well as the balance measurements. The control group did not participate in the training between the pre-test and post-test.

After being assigned to one of the groups, the participants were informed about the research procedure, and they made an appointment for a pre-test. The post-test took place 4 weeks after the pre-test in all training groups and the control group. In the experimental groups, the meetings (10 training sessions for the CT and MT groups, 12 training sessions for the CM group, 3 times a week for 30 min) took place between the pre-test and the post-test training. A visual illustration of the training protocol has been provided in Figure 1. The difference in the number of the training sessions was due to the fact that the subjects in the CT and MT groups did the tasks faster compared to the subjects in the CM group.

Group 1: in the dual-task training (CM), the participants planned their path through the maze and used body balancing on the posturographic platform to move through the maze.

Group 2: in the cognitive training (CT), the participants planned their path through the maze and used a computer keyboard to move through the maze.

Group 3: in the motor training (MT), the participants were to follow the route in the maze along the marked path, using body balancing on the posturographic platform to move through the maze.

Control group: the participants not performing any exercises, neither cognitive nor motor.

All participants, regardless of the training group, worked on the same cognitive material. In order to make the collected data as objective as possible, both the pre-test and the post-test were conducted by a project manager assistant other than the one responsible for the training sessions, and the project manager’s assistants had no insight into the pre-test or the training results.

### 2.3. Research Apparatus

Two DELL Inspiron (500 series) laptops (DELL LLC, Austin, TX, USA) with a 17.3″ screen and 2 Nintendo Wii Balance Board (Nintendo, Kioto, Japan) posturographic platforms of 30 × 50 cm, integrated for the OpenBCI services, were used in all training sessions. Moreover, an open-source DynamicCognition training on the basis of OpenBCI brain–computer interference platform (GNU GPL v3.0 license), operating with Ubuntu 14.04 software, was used. It is designed to improve planning and switching of attention during balance control exercises.

The game consisted of 388 mazes (of increasing difficulty level of cognitive and motor skills) divided into 8 levels of difficulty—higher levels were characterized by a greater number of steps needed to reach the goal. Each maze board was made of 64 square areas (8 × 8). Participants were to navigate through mazes using a green ball, controlled, depending on the experimental group, either by the deviation of the feet pressure center on the posturographic platform (Nintendo Wii Balance Board) (CM and MT group) or by a computer keyboard (CT group). The participants’ task was to move the ball in such a way as to reach the green cross marking the end of the maze, without falling into the area of the “black hole” that would restart the level. After falling into the “black hole” three times, the player went back to the lower level of the task. When moving around the maze, the rules for the ball movement had to be taken into account: the ball moved in a straight line, and it moved until it hit an obstacle (walls, holes, or a cross).

### 2.4. Measures

The Mini-Mental State Examination [24] (the Polish adaptation and standardization) was used for screening the general cognitive ability. The test examines basic cognitive abilities grouped into 6 areas: orientation in time and place, memorization, attention and counting, recalling, language functions, construction praxis. Twenty-four points is the cut-off point that may suggest the dementia process. The higher the MMSE score, the better the performance.

#### 2.4.1. Training Indicators

Special indicators that measured the interference effect were elaborated for the purpose of this study. The game that was created as part of this study allowed for collecting the orientation and planning time and the number of errors data.

##### Time of Orientation and Planning

In all three training groups, this indicator was related to the same time interval in seconds: from the moment the board was displayed until the first step was performed. However, due to the specificity of the training tasks, it measured slightly different functions in the CM and CT groups than in the MT group. In the MT group, the participants were to follow the route in the maze, paying attention to the marked path. In this group, the time of orientation and planning was therefore an indicator of the time needed to get to know the board—of focusing attention and of visual-spatial orientation. In the CM and CT groups, it was an indicator of the time needed to prepare for the task (as in the MT group) and plan the path through the maze. The material on which the participants worked (mazes) was the same for the CM, CT, and MT groups. In the study, the orientation and planning time was averaged for each participant and for all difficulty levels and presented separately for the three training groups.

##### Number of Errors

For the participants in the CM and CT groups, this indicator applies to a slightly different construct than in the MT group, due to the specificity of the training tasks. In the CM and CT groups, the number of errors actually indicated the errors made in the planning process, namely incorrect steps. Incorrectly performed steps are additional moves—they were counted in the form of the difference between all the steps taken in a given maze and the minimum number of steps needed to complete the maze with the correct planning of the path. Planning skills were not tested in the MT group—this indicator concerned the number of deviations from the marked path that had to be followed by tilting the body on the posturographic platform. This error may have been caused by movement or attention difficulties in seniors and was counted each time the participant leaned out on the posturographic platform and kept leaning in a different direction than the path indicated. In the study, the number of errors was averaged for each participant and all levels of difficulty, separately for each of the training groups.

### 2.5. Statistical Analyses

In order to establish whether the data sets had a normal distribution, the Shapiro–Wilk test was used with the significance level *p* < 0.05 [25] corrected for Bonferroni’s multiple comparisons *p* = 0.0125 [26]. The test indicated the lack of normal distribution for the obtained results. Therefore, further analysis was carried out using nonparametric tests.

First, the results for orientation and planning time and the number of errors indicators between the training groups were compared for 8 levels of difficulty using the *U*- Mann–Whitney–Wilcoxon test [27], with the significance level *p* < 0.05 and corrected for Bonferroni’s multiple comparisons *p* = 0.0083 [26].

Secondly, the *W*-Wilcoxon test [28] with the significance level of *p* < 0.05 and corrected for Bonferroni’s multiple comparisons *p* = 0.0083 [26] was used for the comparison of orientation and planning time and the number of errors indicators within all training groups (cognitive-motor, cognitive, motor) on the consecutive 8 levels of difficulty.

To compare the orientation and planning time and the number of errors between all training groups and during the whole training, the *U* Mann–Whitney–Wilcoxon test [26] with the significance level *p* < 0.05 and corrected for Bonferroni’s multiple comparisons *p* = 0.0083 [26] was applied.

In the end, the orientation and planning time and the number of errors were correlated using the *rho*-Spearman (r) [29] with the significance level of *p* < 0.05. Appropriate implementation from Python (version 2.7., SciPy version 0.14.0 an open-source library) was used to calculate the statistics.

## 3. Results

### 3.1. Orientation and Planning Time

In the cognitive-motor and cognitive groups, the plot line had similar shapes compared to the single-motor group (Figure 2). The orientation and planning time successively increased to the 5th level and afterwards, it started to decrease. In the MT group, the subjects needed more time at the beginning of the training. After three levels, the required time was quite short (about 1 s) and kept on the same level till the end of the training. The variety of results was smaller in the CM group compared to the CT group on most levels except the last one. In the MT group, the variety of results was rather small, the highest was observed at the beginning and after that, it remained stable during the rest of the training.

The data in Table 2 shows that the participants in the CT group spent most of the time planning the path compared to other groups in most levels except the last level. The participants who trained in the CM group needed more time for orientation and planning than the elderly in CT and MT groups.

The data in Table 3 show that the subjects in the CM group needed statistically significantly more time for orientation and planning the path on mazes from the 1st to the 5th level. The orientation and planning time was shorter after 6 levels. The participants in the CT group obtained corresponding results to the CM group. The subjects from the MT group needed more time only on the 1st level. Starting from the 2nd level, the required time did not change until the end of the training.

### 3.2. Number of Errors

The results shown in Figure 3 illustrate the similar dynamics of changes after cognitive-motor dual-task and cognitive single-training. The average number of errors increased to the 5th level, and after that, it gradually decreased till the end of the training in both groups. In a single-motor training group, the movement error meant going off the designed path. In MT groups, the errors usually appeared at the beginning of the training and after two levels, they successively decreased. In the CM group, the variety of results was constant, except for the 5th level, where the differentiation was lower. In the cognitive-motor training group, the variety of results was greater than in the cognitive single-task training group, except for the 3rd and 4th level. At these levels, the variation in the data in the CT group was much greater than in the CM group. In the MT group, the differentiation of the results was lower than in the CM and CT groups and it remained on a similar level throughout the whole training.

The results shown in Table 2 reveal that the participants in the CM group made the biggest number of errors compared to the CT and MT groups. Significant differences were especially noticeable when comparing the CM to CT and the CM to MT groups. The subjects who practiced in the CT group made similar errors to the MT group. Participants in the CT groups made more mistakes compared to the MT group on the 1st level. However, the task was difficult for participants in the CT group on the 3rd, 4th, and 7th level compared to the MT group.

The results shown in Table 3 indicate more errors from the beginning of the training to the 5th level. The significant reduction in the number of errors was obtained from the 6th to the 8th level. For participants in the CT group, an increasing number of errors was observed from the 1st level to the 5th level. From the 6th level, a decrease in the number of errors was noted. In the MT group, the number of errors remained constant and from the 4th level, a significant reduction in the number of errors was achieved.

### 3.3. Orientation and Planning Time and the Number of Errors in General

The data in Table 4 suggest that the longest time for orientation and planning was required by the participants in the CT group compared to the CM and MT groups. The largest number of errors was made by participants in the CM group compared to the CT and MT groups.

The correlation data in Table 5 indicate that the orientation and planning time was strongly related to the number of errors in all groups. If the participants needed more time for orientation and planning, they also made more errors during the whole training. In the CM (r = 0.74, *p* < 0.04) and MT (r = 0.90, *p* < 0.002) group, strong relationships were obtained. Only in the CT group, the correlation was strong, but it was at the level of statistical trend (r = 0.70, *p* < 0.06).

## 4. Discussion

The study is of particular importance in the current literature because it shows the mechanism of reduction of the interference effect during dual-task training when compared to the single-task training. The novelty of this study is the analysis of the progress of dual-task and single-task training using two created indicators: orientation and planning time and number of errors. The results allow for a better understanding of the interference effect reduction mechanisms in the elderly. There were two general assumptions made on the basis of the literature, (1) that the planning time would be longer, and the number of errors would be larger during the dual-task training and that (2) the planning time would be shorter, and the number of errors would be reduced after all types of training (dual and single-tasks training).

### 4.1. The Planning Time Will Be Longer, and the Number of Errors Will Be Larger during the Dual-Task Training, Compared to the Single-Task Training

The obtained results have shown that the training in a dual-task condition was associated with a significantly greater number of errors, but not with a longer task planning time compared to a single-task condition training.

The results of this study visibly differ from the classic studies about the Stroop interference effect [13,14,15,16,17]. In the studies based on the Stroop task paradigm, the elderly participants obtained a longer executive time and made more errors in interference color-word blocks compared to non-interference blocks. It is noteworthy that in this study, the time of task planning was analyzed, and in the discussed works, the actual time of task execution was considered. These are not the same variables. Due to technical reasons, it was impossible to analyze the time needed to complete the paths through the mazes in the three examined training groups. Another difference is the fact, that the Stroop interference tasks cannot be planned before their execution. Participants must react to the emerging stimuli. In this study, the subjects in the CM and CT training groups were given instructions and they were asked first to plan the task and then to execute it. In the cognitive-motor group, the participants had to plan the path, keep it in their working memory, and execute it using swings on the posturographic platform. Perhaps too many tasks were performed simultaneously, and they did not fit into the working memory capacity of the subjects. To reduce the interference effect in the CM group, the participants planned the whole path not at the beginning (according to the instruction), but they planned it after every few moves. The subjects in the CT group were supposed to plan the path and complete it by pressing arrows on the computer keyboard. The subjects performing only one operation at a time could plan the entire path at the beginning and keep it in their working memory.

### 4.2. The Planning Time Will Be Shorter, and the Number of Errors Will Be Reduced after All Types of Training

There was a decrease in the time needed to plan a path in the mazes among the subjects training in the cognitive-motor, cognitive, and motor groups. From the 6th level of difficulty, an increase of the required time in cognitive-motor and cognitive tasks was obtained. In the MT group, a significant reduction in the time needed to start a task was obtained on the 2nd level. There was a pronounced reduction in the number of errors made in the cognitive-motor, cognitive, and motor training. The subjects who performed cognitive-motor and cognitive exercises made fewer errors from the 6th level of difficulty. The participants training the motor tasks less frequently got off the path from the 3rd difficulty level. This result means that after all types of training, the number of errors and the planning time are reduced, but it happens at different moments during the training.

These results are consistent with the current literature, where in tasks based on the Stroop interference effect, the elderly subjects obtained shorter execution times and made fewer errors [8,9,10,11]. Relating the results to the theory, the subjects learn to perform the task during cognitive-motor, cognitive, and motor training. Perhaps, following Kahneman’s [5,6] attention resources theory, cognitive and movement task balances the resources of attention in the elderly. Therefore, they make more mistakes than the respondents from the two single-training groups. In the process of learning, the task is automated, and the participants begin to make fewer mistakes and plan the route faster.

The results indicate that after all types of training, the number of errors and the time needed to plan the path decreases, despite the increasing difficulty of the tasks. Depending on the type of training, the planning time and errors are reduced at different points in the training.

### 4.3. General Discussion

In comparison to the neuropsychological studies mentioned above, the dual-task training presented in this paper was more cognitively demanding. The interference effect was visible, especially at the beginning of the training and it was present up to the 6th difficulty level, after that it decreased. In the context of future research, an interesting one seems to be assessing the similarities of the learning effect for cognitive-motor and cognitive training. Perhaps the balance training was of a slight difficulty for the participants who practiced cognitive-motor tasks and the cognitive-motor and cognitive groups in fact worked on the same skills. Additionally, no significant effect of exercise training in the motor group was demonstrated. On the other hand, in the dual-task training, an increased number of errors was observed, and even a slight “strain” with the task that followed caused an interference.

The obtained results will have significant applicability in practice. The dual-task training could be one of highly effective, additional methods of supporting older adults in terms of their cognitive and physical activity. What is more, it may become a possible means of providing rehabilitation, thus leading to maintaining their independence in everyday life.

### 4.4. Limitation of the Study

Several limitations of the study should be considered. One of these limitations is the sample blinding method. In the discussed paper, the double-blind standard was not maintained. The project assistants knew to which groups the participants were assigned. Another limitation is the cognitive material (the maze game with increasing difficulty) used in the training sessions could be repetitive and monotonous for participants, possibly implicating reduced motivation. The last limitation would be the participants group. The participants were a group of healthy older adults, mostly women (73% of participants), seeking some cognitive activity.

## 5. Conclusions

Based on the collected data, it can be concluded that dual-task cognitive-motor training is more cognitively demanding than a training to perform single activities, which is visible concerning some of its characteristics. It manifests in a greater number of errors made throughout the training, but it does not extend the orientation and planning time compared to cognitive single-task training.

## Figures and Tables

**Figure 1 ijerph-20-01477-f001:**
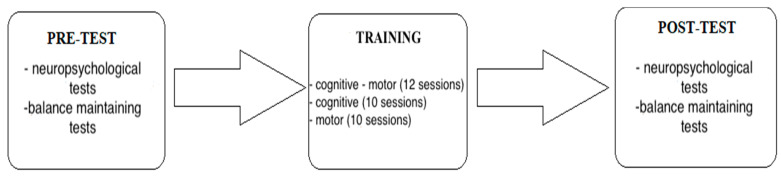
General scheme of the study.

**Figure 2 ijerph-20-01477-f002:**
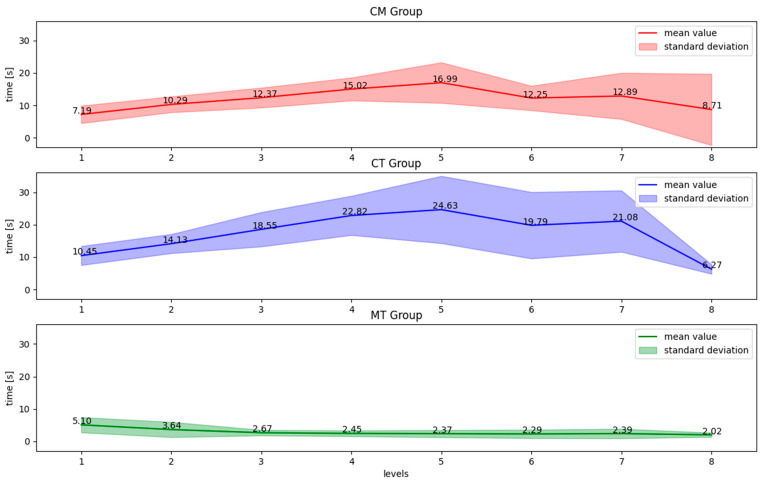
Average orientation and planning time for each difficulty level in all training groups: the **top** chart—dual-task cognitive-motor group, the **middle** chart—single training cognitive group, the **bottom** chart—single training motor group.

**Figure 3 ijerph-20-01477-f003:**
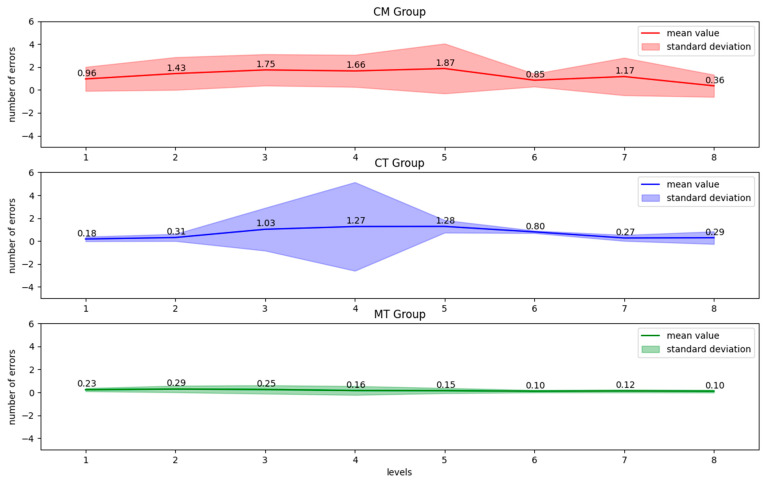
Average Number of Errors for each difficulty level in all training groups: the **top** chart—dual-task cognitive-motor group, the **middle** chart—single training cognitive group, the **bottom** chart—single training motor group.

**Table 1 ijerph-20-01477-t001:** Demographic and clinical variables in all groups.

	CM M (SD)	CT M (SD)	MT M (SD)	C M (SD)	*p*-Value ANCOVA
Age	71.7 (5.0)	72.4 (5.4)	70.1 (6.2)	70.5 (4.1)	0.413
Years of education	15.8 (2.0)	15.2 (2.9)	15.1 (2.2)	14.9 (2.9)	0.983
MMSE	29.0 (1.0)	28.9 (1.0)	29.4 (0.7)	29.1 (1.0)	0.975
Gender	F—14 M—5	F—17 M—3	F—13 M—6	F—16 M—4	0.757

Abbreviations: CM—cognitive-motor training group, CT—cognitive training group, MT—motor training group, C—control group, M—mean, SD—standard deviation, MMSE—Mini-Mental State Examination, F—female, M—men.

**Table 2 ijerph-20-01477-t002:** Orientation and planning time and the number of errors comparison for all levels.

Level		1	2	3	4	5	6	7	8
Orientation and planning time ^1^	CM M (SD)	7.19 (2.70)	10.29 (2.43)	12.37 (3.07)	15.02 (3.52)	16.99 (6.25)	12.25 (3.77)	12.89 (7.12)	8.71 (10.99)
	CT M (SD)	10.45 (2.93)	14.13 (2.94)	18.55 (5.29)	22.82 (6.04)	24.63 (10.36)	19.79 (10.25)	21.08 (9.46)	6.27 (1.45)
	MT M (SD)	5.10 (2.33)	3.64 (2.34)	2.67 (0.85)	2.45 (0.88)	2.37 (1.11)	2.29 (1.30)	2.39 (1.47)	2.02 (0.66)
	*U* CM-CT (*p* < 0.05)	74.0 (0.01)	60.0 (0.01)	61.0 (0.01)	51.0 (0.001)	97.0 (0.01)	56.0 (0.01)	56.0 (0.01)	36.0 (0.24)
	*U* CM-MT (*p* < 0.05)	87.0 (0.01)	14.0 (0.001)	0.0 (0.001)	0.0 (0.001)	0.0 (0.001)	0.0 (0.001)	0.0 (0.001)	3.0 (0.001)
	*U* CT-MT (*p* < 0.05)	33.0 (0.001)	6.0 (0.001)	0.0 (0.001)	0.0 (0.001)	0.0 (0.001)	0.0 (0.001)	5.0 (0.001)	0.0 (0.001)
Number of errors	Group CM M (SD)	0.96 (1.05)	1.43 (1.43)	1.75 (1.37)	1.66 (1.40)	1.87 (2.18)	0.85 (0.56)	1.17 (1.64)	0.36 (0.97)
	Group CT M (SD)	0.1.8 (0.20)	0.31 (0.31)	0.25 (0.37)	0.16 (0.39)	0.15 (0.23)	0.1 (0.11)	0.12 (0.13)	0.1 (0.12)
	Group MT M (SD)	0.23 (0.14)	0.29 (0.29)	0.25 (0.37)	0.16 (0.39)	0.15 (0.23)	0.1 (0.11)	0.12 (0.13)	0.1 (0.12)
	*U* CM-CT (*p* < 0.05)	54.5 0.001	53.0 0.001	77.0 0.001	57.5 0.001	41.0 0.001	25.0 0.001	47.5 0.001	24.0 0.05
	*U* CM-MT (*p* < 0.05)	69.0 0.001	46.0 0.001	23.0 0.001	12.0 0.001	18.0 0.001	25.0 0.001	27.0 0.001	38.0 0.001
	*U* CT-MT (*p* < 0.05)	131.0 0.05	179.0 0.49	84.0 0.001	105.0 0.001	139.0 0.17	137.5 0.31	86.5 0.02	52.0 0.25

^1^ Average time in s. Abbreviations: CM—cognitive-motor training group, CT—cognitive training group, MT—motor training group, M—mean, SD—standard deviation, *U*—Mann–Whitney–Wilcoxon test, *p*—significance level.

**Table 3 ijerph-20-01477-t003:** Orientation and planning time and the number of errors comparison between different difficulty levels in all groups.

Difficulty Levels Compared		1–2	2–3	3–4	4–5	5–6	6–7	7–8
Orientation and planning time	CM *W* (*p*)	56.0 (0.01)	103.0 (0.01)	108.0 (0.01)	160.0 (0.28)	89.0 (0.01)	142.0 (0.36)	31.0 (0.01)
	CT *W* (*p*)	79.0 (0.01)	103.0 (0.01)	108.0 (0.01)	182.0 (0.41)	101.0 (0.02)	120.0 (0.29)	1.0 (0.01)
	MT *W* (*p*)	82.0 (0.01)	115.0 (0.07)	127.0 (0.13)	135.0 (0.20)	138.0 (0.22)	147.0 (0.32)	133.0 (0.18)
Number of errors	CM *W* (*p*)	138.0 (0.11)	151.0 (0.19)	170.0 (0.38)	175.5 (0.45)	107.0 (0.02)	144.0 (0.39)	72.0 (0.05)
	CT *W* (*p*)	131.5 (0.03)	135.5 (0.04)	138.0 (0.05)	152.0 (0.15)	131.0 (0.17)	76.0 (0.02)	36.0 (0.1)
	MT *W* (*p*)	169.0 (0.49)	130.0 (0.16)	107.0 (0.05)	136.5 (0.21)	156.0 (0.43)	153.0 (0.39)	131.0 (0.16)

Abbreviations: CM—cognitive-motor training group, CT—cognitive training group, MT—motor training group, *W*—Wilcoxon test, *p*—significance level.

**Table 4 ijerph-20-01477-t004:** Orientation and planning time and the number of errors comparison during the whole training.

Indicator	CM M (SD)	CT M (SD)	MT M (SD)	*U* (*p* < 0.05) CM-CT	*U* (*p* < 0.05) CM-MT	*U* (*p* < 0.05) CT-MT
Orientation and planning time	95.71 (6.14)	137.72 (8.82)	22.92 (1.79)	15.0 (0.04)	0.0 (0.01)	0.0 (0.01)
Number of errors	1.26 (1.71)	0.46 (1.71)	0.17 (0.25)	7.0 (0.01)	0.0 (0.01)	14.0 (0.03)

Abbreviations: CM—cognitive-motor training group, CT—cognitive training group, MT—motor training group, M—mean, SD—standard deviation, *U*—Mann–Whitney–Wilcoxon test, *p*—significance level.

**Table 5 ijerph-20-01477-t005:** Rho-Spearman ^1^ correlation between the orientation and planning time and the number of errors during the whole training.

Training Group	r Test	*p* < 0.05
CM	0.74	0.04
CT	0.70	0.06
MT	0.90	0.002

^1^ The strength of the absolute correlation value is: 0.00–0.19 “very weak”, 0.20–0.39 “weak”, 0.40–0.59 “moderate”, 0.60–0.79 “strong”, 0.80–1.0 “very strong” (King, Minium, 2004). Abbreviations: CM—cognitive-motor training group, CT—cognitive training group, MT—motor training group, r—Rho-Spearman correlation, *p*—significance level.

## Data Availability

Not applicable.

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
