# Peer review of "The Characteristics of the Reduction of Interference Effect during Dual-Task Cognitive-Motor Training Compared to a Single Task Cognitive and Motor Training in Elderly: A Randomized Controlled Trial"

_ijerph, 2023, doi:10.3390/ijerph20021477_

Round 1
Reviewer 1 Report
Comments to manuscript ID IJERPH-2068413
In this work, the authors intend examining the characteristics of dual-task training compared to a single task cognitive or motor training in elderly.
This study is well-written, adequately organized and the structure of the randomized controlled trial has been well-planned and conducted. However, some major concerns should be raised.
In the “materials and methods” section the total amount of screened participants (enrolled/not enrolled) is not clearly specified, considering the quite perfect splitting in three homogeneous groups.
In “Randomization” section the authors declare that each examined person was randomly assigned to one of four groups and mentioned a Control Group, but in the text there is no statement about the characteristic of this group and its role in the trial.
I think that authors have correctly chosen MMSE as a screening test for eligibility criteria, but if they want to correlate the role of interference effect in the elderly they should use a more sensitive test for executive functions.
The “results” section is adequately described and the number of tables is sufficient to describe all the data obtained.
Regarding the discussion section, the authors should emphasize the importance of the trial and data obtained, rather than simply summarized what they have found. The novelty of each research article is given by opening hypotheses and correlations underlying good or even bad study results. Moreover, they should highlight the importance of a possible “rehabilitative” role using a dual-task training even in elderly, as already carried out with sick people.
Author Response
Dear Sir or Madam,
Thank you for all your suggestions to help increase the impact of this paper.
The reviewers’ comments and suggestions have been very helpful and certainly all the changes made in the manuscript have enhanced its quality. In reference to the reviewers’ comments,
the following changes in the article have been made:
- In the “materials and methods” section the total amount of screened participants (enrolled/not enrolled) is not clearly specified, considering the quite perfect splitting in three homogeneous groups.
The total amount of participants was clarified in the “materials and methods” section. The demographic information about the control group was added to Table 1.
- In “Randomization” section the authors declare that each examined person was randomly assigned to one of four groups and mentioned a Control Group, but in the text there is no statement about the characteristic of this group and its role in the trial.
The information about the control group was added in the “materials and methods – procedure” section.
- I think that authors have correctly chosen MMSE as a screening test for eligibility criteria, but if they want to correlate the role of interference effect in the elderly they should use a more sensitive test for executive functions.
In this study 2 times (pre-test and post-test) the neuropsychological assessment (executive functions and attention) was conducted. The MMSE was only a screening and scored 24 points or more were an included criterium. The information about neuropsychological assessment was added to the “materials and methods – procedure” section.
I attach the list of neuropsychological tests:
- The Color Trails Test for Adults (CTT) - (D'Elia et al. 1996)
2.The Ruff Figural Fluency Test (RFFT) - (Ruff et al., 1987)
3.The Wisconsin Card Sorting Test (WCST) - (Grant, Berg, 1948)
4.The Digit Span from the Wechsler Intelligent Adults Scale (DS) - (Wechsler, 1997)
5.The Verbal Fluency Test (VF) - (Benton et al., 1994)
6.The Stroop Color Word Test (SCWT) – (Stroop, 1935)
- Regarding the discussion section, the authors should emphasize the importance of the trial and data obtained, rather than simply summarized what they have found. The novelty of each research article is given by opening hypotheses and correlations underlying good or even bad study results. Moreover, they should highlight the importance of a possible “rehabilitative” role using a dual-task training even in elderly, as already carried out with sick people.
The needed information in the discussion section was added.
Reviewer 2 Report
In this manuscript, the authors investigated the effect of dual task cognitive motor training on the interference effect in the elderly. The research question is important and the study is generally well conducted. The reviewer, however, has several suggestions that may help improve the manuscript.
1, abstract: please add some background information in the beginning to put the study into context. Regarding the description of results, can the authors report detailed statistics such as correlation coefficients and other effect size measures to let the readers understand the size of the observed effects?
2, introduction: the first sentence needs references.
Can the authors provide more recent citations for the limited resources and bottleneck theories? preferably recent reviews.
lines 65-66: the description of fMRI studies needs references.
lines 82-85: "planning time" and "number of errors" are not mentioned before the assumptions, can the authors clarify their relation with the interference effect? what do they measure?
3, methods: why the authors set the sample size to 65, did they conduct a priori power analysis?
Can the authors provide a visual illustration of the training protocol for each group? a figure may greatly facilitate the understanding of the interventions.
For the statistical analysis, did the authors first test the normal distribution of the data? parameter tests are generally preferred to nonparametric ones if the assumptions are met.
4, results: table 1, why ANCOVA? it seems simple ANOVA is sufficient.
Table 5, can the authors provide a scatterplot to show the correlations are genuine?
Author Response
Dear Sir or Madam,
Thank you for all your suggestions to help increase the impact of this paper.
The reviewers’ comments and suggestions have been very helpful and certainly all the changes made in the manuscript have enhanced its quality. In reference to the reviewers’ comments,
the following changes in the article have been made:
- abstract: please add some background information in the beginning to put the study into context. Regarding the description of results, can the authors report detailed statistics such as correlation coefficients and other effect size measures to let the readers understand the size of the observed effects?
The information was added in the abstract section.
- introduction: the first sentence needs references.
Can the authors provide more recent citations for the limited resources and bottleneck theories? preferably recent reviews.
lines 65-66: the description of fMRI studies needs references.
All references were added in the suggested places.
lines 82-85: "planning time" and "number of errors" are not mentioned before the assumptions, can the authors clarify their relation with the interference effect? what do they measure?
Information about Orientation and planning time and the Number of errors were clarified in the ‘materials and methods” section and “introduction” section based on interference effect theory.
- methods: why the authors set the sample size to 65, did they conduct a priori power analysis?
The total amounts of screened participants were 84 subjects. The information was added in the “materials and methods” section. The a priori power analysis has not been performed in this study.
Can the authors provide a visual illustration of the training protocol for each group? a figure may greatly facilitate the understanding of the interventions.
A visual illustration was added in the “results” section.
For the statistical analysis, did the authors first test the normal distribution of the data? parameter tests are generally preferred to nonparametric ones if the assumptions are met.
The additional information about the first test the normal distribution was added in the “materials and methods – statistical analysis” section.
4) results: table 1, why ANCOVA? it seems simple ANOVA is sufficient.
The data does not allow the assumptions of normal distribution.
Table 5, can the authors provide a scatterplot to show the correlations are genuine?
A scatterplot with correlation was added in the “results” section.
Round 2
Reviewer 1 Report
I think the paper is much improved and that the authors answered adequately to reviewers’ suggestions. I don't have further remarks
Author Response
Thank you very much for your opinion and all the earlier suggestion.
Reviewer 2 Report
Thank the authors for addressing my concerns.
1), please restructure the abstract, the new results added at the end should be moved to the results part, not the end.
2), so how did the author determine the sample size of the study? what did the six subjects drop out?
3), regarding the control group, it seems that the authors did not present any data, so what is the purpose of including this control group?
4), I cannot find any information on planning time and number of errors in the introduction, can the authors show me the exact place where this information is added?
Author Response
Dear Sir or Madam
Thank you for your all suggestion.
1), please restructure the abstract, the new results added at the end should be moved to the results part, not the end.
The abstract was restructured, and the new results were added to the results part.
2), so how did the author determine the sample size of the study? what did the six subjects drop out?
At the research planning stage, we assumed 20 participants in each training and control group. It is related usually to the minimal number of results needed to use many statistical tests.
The information about the dropped-out reason was added in the Materials and Methods section.
3), regarding the control group, it seems that the authors did not present any data, so what is the purpose of including this control group?
The other Reviewer clearly emphasized the necessity of adding the control group information. That’s why the information appeared in the last manuscript version.
4), I cannot find any information on planning time and number of errors in the introduction, can the authors show me the exact place where this information is added?
In the introduction, we added the theoretical background of created indicators: Orientation and planning time and Number of Errors. “The completion time of dual-task, in comparison to the single-task, increases and more errors may appear; moreover, considerable attention resources and memory, thinking, planning, which are components of EF are also used (58-61 lines)”.